

# Acute responses and recovery in the femoral cartilage morphology following running and cool-down protocols

Sanghyup Park[1], Junhyeong Lim[1], Jinwoo Lee[1], Seonggyu Jeon[1], Jaewon Kim[2] and Jihong Park[2]

[1] Department of Physical Education, Graduate School, Kyung Hee University, Yongin, Republic of Korea
[2] Department of Sports Medicine, Kyung Hee University, Yongin, Republic of Korea

## ABSTRACT

This study compared the immediate effects of two common post-exercise cool-down methods to a control condition on subsequent morphologic change in femoral cartilage and vascular response in the posterior tibial artery after running. Sixteen healthy young males (23.6 ± 2.2 years, 172.8 ± 4.9 cm, 72.2 ± 7.1 kg) visited the laboratory during three separate sessions and performed 30-min of treadmill running (7.5 km/h for the initial 5-min, followed 8.5 km/h for 25-min). After running, participants experienced one of three 30-min cool-down protocols: active cool-down, cold application, or control (seated rest with their knee fully extended), in a counterbalanced order. Ultrasonographic assessments of femoral cartilage thickness (intercondylar, lateral, and medial) and posterior tibial artery blood flow were compared. To test condition effects over time, two-way analysis of variances and Tukey tests were used ($p < 0.05$) with Cohen's d effect sizes (ES). There was no condition by time interaction in femoral cartilage thickness (intercondylar: $F_{30,705} = 0.91$, $p = 0.61$; lateral: $F_{30,705} = 1.24$, $p = 0.18$; medial: $F_{30,705} = 0.49$, $p = 0.99$). Regardless of time (condition effect: $F_{2,705} > 3.24$, $p < 0.04$ for all tests), femoral cartilage in the cold application condition was thicker than the control condition (intercondylar: $p = 0.01$, ES = 0.16; lateral: $p < 0.0001$, ES = 0.24; medial: $p = 0.04$. ES = 0.16). Regardless of condition (time effect: $F_{15,705} > 10.31$, $p < 0.0001$ for all tests), femoral cartilage thickness was decreased after running (intercondylar: $p < 0.0001$, ES = 1.37; lateral: $p < 0.0001$, ES = 1.58; medial: $p < 0.0001$, ES = 0.81) and returned to baseline levels within 40-min (intercondylar: $p = 0.09$; lateral: $p = 0.64$; medial: $p = 0.26$). Blood flow volume was different (condition × time: $F_{30,705} = 2.36$, $p < 0.0001$) that running-induced blood flow volume was maintained for 30-min for the active cool-down condition ($p < 0.0001$, ES = 1.64), whereas it returned to baseline levels within 10-min for other conditions (cold application: $p = 0.67$; control: $p = 0.62$). Neither blood flow nor temperature had a significant impact on the recovery in femoral cartilage after running.

Corresponding author
Jihong Park, jihong.park@khu.ac.kr

## INTRODUCTION

Articular cartilage, described as avascular and aneural connective tissue of diarthrodial joints, comprises 60~80% fluid content at rest (*Lad et al., 2016*). Its primary function includes absorbing and distributing biomechanical forces generated during daily activities and exercises (*Oinas et al., 2018*). Under normal hydrated condition, interstitial fluid and pressure resists mechanical loading, thereby preventing cartilage friction and strain (*Ateshian, 2009*). Continual mechanical loading leads to the dispersion of interstitial pressure and exudation, resulting in morphologic deformation (*Halonen et al., 2014*). Upon cessation of physical activity, the exudated fluid is gradually reabsorbed over time (*Lad et al., 2016*). While several mechanisms of cartilage fluid retention have been proposed, the competing rates between fluid exudation and recovery appears to be a determining factor (*Voinier et al., 2022*).

The ultrasonographic assessment has been validated and deemed reliable (*Schmitz et al., 2017*) for imaging lower extremity cartilage morphology (*Lee et al., 2023*; *Song et al., 2020*). Cartilage deformation associated with weight bearing and its subsequent recovery typically occur within hours (*Harkey et al., 2018*). Ultrasonography is an efficient imaging technique in athletic settings, allowing for repetitive measurements. Recent ultrasonographic studies reported thickness reductions of 3% (*Lee et al., 2023*), 6% (*Lim et al., 2024*), and 9% (*Harkey et al., 2017*) in femoral cartilage after treadmill running. Upon cessation of exercise, it took 30- and 45-min for deformed femoral cartilage to return to pre-exercise levels under conditions of static unloading (*Harkey et al., 2018*). Previous data suggest that both time (*Cutcliffe et al., 2020*) and intensity (*Eckstein et al., 2005*) play significant roles in physical activity-associated cartilage deformation. Additionally, the recovery rate seems to be proportional to the magnitude of deformation (*Harkey et al., 2018*).

While the idea of dose-dependent factors such as time and intensity affecting acute deformation in articular cartilage has been well-documented (*Cutcliffe et al., 2020*; *Harkey et al., 2018*), less data are available regarding the recovery rate after physical activity, such as running. While cumulative joint loading from repetitive or prolonged weight bearing activities may lead to pathological conditions (*Harkey et al., 2018*), understanding the duration required for cartilage to return to its normal shape after a single bout of exercise could provide clinical insight into the biomechanics of cartilage recovery and long-term joint health. Furthermore, the influence of common post-exercise cool-down strategies on the recovery process of exercise-induced cartilage deformation is unknown. In the field of sports medicine, active cool-down (*Van Hooren & Peake, 2018*) and cold application (*Lim et al., 2022*) are commonly recommended. Cartilage hydrodynamics, involving osmotic swelling and articular contact area within a joint (*Voinier et al., 2022*), could be influenced by blood flow (*Moeini, Lee & Quinn, 2012*). As joint temperature increases, the diffusivity of articular cartilage matrices increases and the density of water decreases, potentially influencing cartilage morphology (*Moeini, Lee & Quinn, 2012*). Although articular cartilage is devoid of blood vessels, lymphatic capillaries, and nerves (*Lad et al., 2016*), cool-down methods that alter blood flow or temperature could possibly enhance reabsorption of water into the cartilage.

Hence, the purpose of this study was to compare the effects of active cool-down and cold application to a control condition (*e.g.*, unloading rest with a seated position) on subsequent femoral cartilage thickness following 30-min treadmill running. Given that femoral cartilage thickness varies between regions (*Harkey et al., 2018*), it would be reasonable to separately assess the intercondylar, lateral, and medial thickness. Alongside the evaluation of femoral cartilage morphology, simultaneous assessment of blood flow volume at the posterior tibial artery would address the limitations regarding the unclear effect of post-exercise cool-down interventions on the recovery of femoral cartilage. Therefore, we asked following research questions: Will the recovery rate on exercise-induced deformation in the femoral cartilage following the three cool-down protocols differ? and we hypothesised that exercise-induced femoral cartilage deformation would be recovered faster for the active cool-down and cold application conditions compared to the control condition.

## METHODS

### Study design

A crossover design with repeated measures was used. Participants visited the laboratory at the same time of the day for three separate days, each for three different cool-down protocols (Fig. 1). Each session (with at least 48-h between visits), participants performed each condition in a counterbalanced order. Independent variables were cool-down methods (active cool-down, cold application, and seated rest) and time. Specifically, there were 16 time points: every 5-min during the 30-min unloading phase, post-run, cool-down at 10- and -30-min, and every 5-min during the 30-min post cool-down phase. Dependent measurements included femoral cartilage thickness (intercondylar, lateral, and medial) and posterior tibial artery blood flow volume. The ambient temperature and relative humidity in the laboratory were recorded as 25.3 ± 2.5 °C and 60.1 ± 5.3% during the data collection period.

### Participants

Sixteen healthy young males (23.6 ± 2.2 years, 172.8 ± 4.9 cm, 72.2 ± 7.1 kg) volunteered to participate. To be eligible to the study, participants had to be free of any history of lower-extremity injury in the past 6 months and orthopaedic surgery for their lifetimes. Participants were excluded if they had any neuromuscular disorder, cardiopulmonary diseases, or medical conditions. Prior to participation, all participants provided written informed consent, approved by the Kyung Hee University Institutional Review Board (protocol number: KHGIRB-21-222).

### Testing procedures

#### Unloading phase

Upon arrival to the laboratory, participants read the testing procedures and gave written informed consent during the first visit. Afterwards, participants were seated on a treatment table with their backs against the wall and both knees fully extended for 30-min to remove
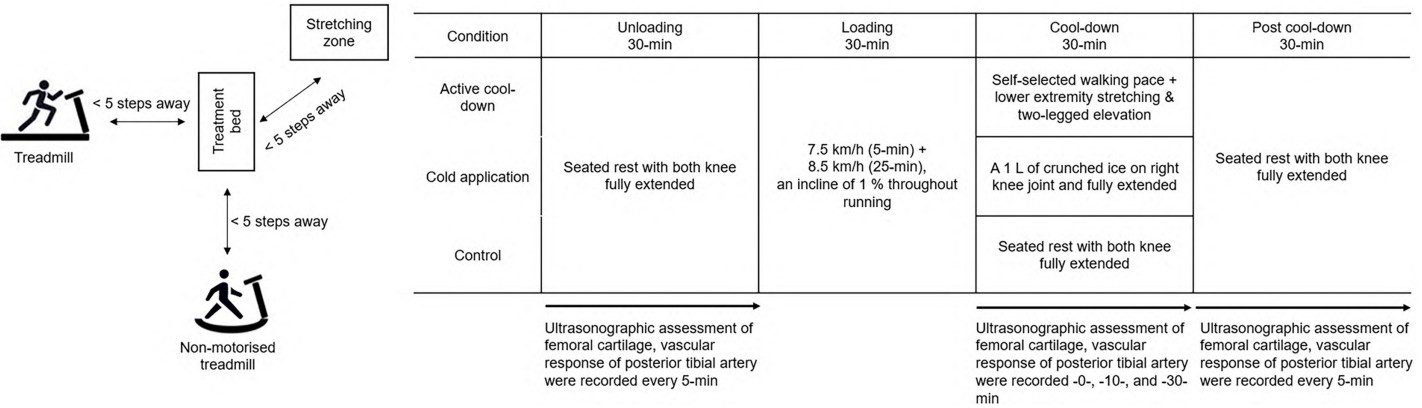

**Figure 1 A depiction of the experimental set up and testing procedures.**

mechanical load from the preceding weight-bearing activity on the femoral cartilage (*Harkey et al., 2018*).

Baseline images of the femoral cartilage were recorded every 5-min using ultrasound imaging devices equipped with a 12 MHz linear transducer (Ecube-i7, Ecube-12, Alpinion, Anyang, Korea). Participants bent their right knee to 140° using a plastic goniometer (*Harkey et al., 2017*), and in this position, the distance between the wall and the right hallux was marked using a tape measure for subsequent assessments. The transducer was then placed transversely in line with the lateral and medial femoral condyles above the superior edge of the patella. A transparency grid was placed over the ultrasound device screen to aid assessment reproducibility (*Harkey et al., 2017*). When sampling images, the intercondylar notch was centred, and the positioning of the lateral and medial condyles at the edge of the transparency grid on the screen was recorded. At each time point, three images were taken.

Baseline values of the posterior tibial artery blood flow were also recorded every 5-min of the unloading phase. The same transducer was positioned 10 cm above from the right medial malleolus and slightly moved posteriorly. Magnification and focal zone setting were first adjusted to optimise imaging of the vessel walls. Frequency (6.0 MHz) and gain (18 dB) settings were maintained throughout the whole data collection sessions. A line corresponding with the long axis of the posterior tibial artery that was most clearly viewed was measured to insert the diameter (*Sabatier et al., 2006*). At the same time, a built-in software programme in the ultrasound devices, using the formula: Blood flow volume = diameter$^2$ × 0.785 × time-averaged velocity (*Blanco, 2015*), automatically assessed the blood flow volume at the posterior tibial artery.

### Loading phase

Participants moved to a treadmill (Jog Forma, Technogym, Cesena, ITA) and performed a 30-min run. The running protocol consisted of a belt speed of 7.5 km/h for the initial 5-min, followed by an increase to 8.5 km/h for the next 25-min (an incline of 1% for entire running) each session.

**Table 1 Active cool-down protocol.**

| Order | Activity | Intensity | Time |
|---|---|---|---|
| 1 | Self-selected walk | Self-paced | 10-min |
| 2,6 | Static stretch (hamstrings, gluteal muscles, and quadriceps) | Mild discomfort | 2-min (a 20-s hold for each side) |
| 3,5,7 | Two-legged elevation | | 3-min |
| 4 | Static stretch (gastrocnemius, soleus, hip adductors) | Mild discomfort | 2-min (a 20-s hold for each side) |

Note:
After the completion of 1 (cool-down-10-min) and 7 (cool-down-30-min), ultrasonographic assessments (femoral cartilage and posterior tibial artery blood flow), blood lactate concentration, and fatigue perception were recorded.

### Cool-down phase

Participants returned to the treatment table and experienced one of three 30-min cool-down methods in a counterbalanced order: (1) Active cool-down: Participants performed 10-min of self-paced walking on a non-motorised treadmill (Skillmill, Technogym, Cesena, Italia), followed by three bilateral static stretch (a 20-s hold for each side on hamstrings, gluteal muscles, hip adductors, quadriceps, soleus, and gastrocnemius) and a two-legged elevation on a stretch mat (Table 1). The distance among the treatment table, non-motorised treadmill, and stretch mat was <5 steps (Fig. 1). The intensity of static stretching was controlled to a subjective point at which participants felt mild discomfort and stopped just before it become painful (Takeuchi & Nakamura, 2020); (2) Cold application: An ice bag (1 L of crushed ice in a 30 cm × 50 cm plastic bag, GCspocare, Seoul, Korea) was applied on the right kneecap while participants seated on a treatment table with both knees fully extended. A weighted sandbag (1.8 kg, 12 cm × 33 cm; Sammons Preston, Bolingbrook, USA) was applied on top of the ice bag to increase thermal conductivity; and (3) Control: Participants maintained the unloading position. Each protocol (e.g., stretch exercises, ice bag application, etc.) and duration (using a stopwatch) of cool-down methods were controlled by two pre-trained researchers. Ultrasound images for femoral cartilage and posterior tibial artery blood flow volume were taken at 10- and 30-min of the cool-down phase.

### Post-cool-down phase

After the cool-down methods, participants were returned to the unloading position for 30-min. Ultrasound images for femoral cartilage and posterior tibial artery blood flow were recorded every 5-min thereafter.

### Analysis of femoral cartilage images

Recorded ultrasound images were exported to ImageJ software (National Institutes of Health, Bethesda, USA). The examiner (X) with a within- and between-session reliability (ICC) of 0.87 and 0.83, respectively, analysed all ultrasound images (Shrout & Fleiss, 1979). The information regarding condition and time associated with data acquisition was not shown on each image; thus, the examiner was blinded to these factors. A straight line between the lateral and medial upper edge of the femoral cartilage was drawn, which was aligned parallel to an angle of 0° so each edge was levelled. Lateral and medial lengths were
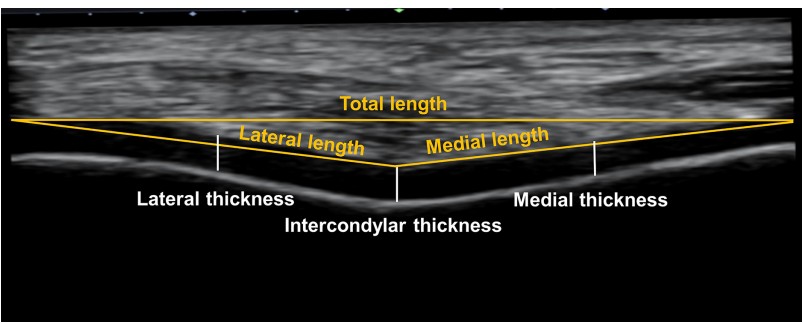

**Figure 2 Analysis of femoral cartilage images.**

then defined as straight lines from the intercondylar notch to the lateral and medial upper edges, respectively (*Song et al., 2020*). The intercondylar thickness was assessed by a vertical straight line from the intercondylar notch to the cartilage–bone interface (*Harkey et al., 2017*). Similarly, the lateral and medial thickness were also assessed by the perpendicular lines at the midpoint of the lateral and medial lengths between the synovial space–cartilage interface and cartilage–bone interface. Three values at each time point across conditions were averaged and analysed (Fig. 2).

## Statistical analysis

Our sample size was determined using an expected change in the medial condyle thickness of the femoral cartilage of 2.61 mm a standard deviation of 0.61 mm (*Harkey et al., 2018*). With an effects size of 0.79, this calculation estimated that 15 participants would be necessary in each condition (an alpha of 0.05 and a beta of 0.8).

Mean and 95% confidence intervals from each dependent measurement were calculated across conditions at each time point. To test condition effect over time, a two-way mixed model analysis of variance was conducted, with the random variable being participant and the fixed variables being condition and time. Tukey-Kramer pairwise comparisons were used as *post-hoc* tests. A statistical package (SAS Ver. 9.4, SAS Institute, Cary, NC, USA) was used for all tests ($\alpha = 0.05$). Effect sizes (ES = [$\bar{X}_1 - \bar{X}_2$] / $\sigma_{pooled}$) were also calculated to determine practical significance (*Cohen, 1962*).

## RESULTS

### Intercondylar thickness

There was no condition effect over time in the intercondylar thickness (condition × time: $F_{30,705} = 0.91$, $p = 0.61$, Fig. 3). Regardless of time (condition effect: $F_{2,705} = 4.02$, $p = 0.02$), intercondylar thickness in the cold application condition was thicker than the control condition (2.20 *vs.* 2.18 mm, $p = 0.01$, ES = 0.16). Regardless of condition (time effect: $F_{15,705} = 49.39$, $p < 0.0001$, Fig. 3), there was an immediate reduction in the intercondylar thickness after running (cool-down-0-min: 2.02 mm, −11%, $p < 0.0001$, ES = 1.37), followed by a return to baseline levels (2.28 mm) at post-cool-down-10-min (2.21 mm, $p = 0.09$).

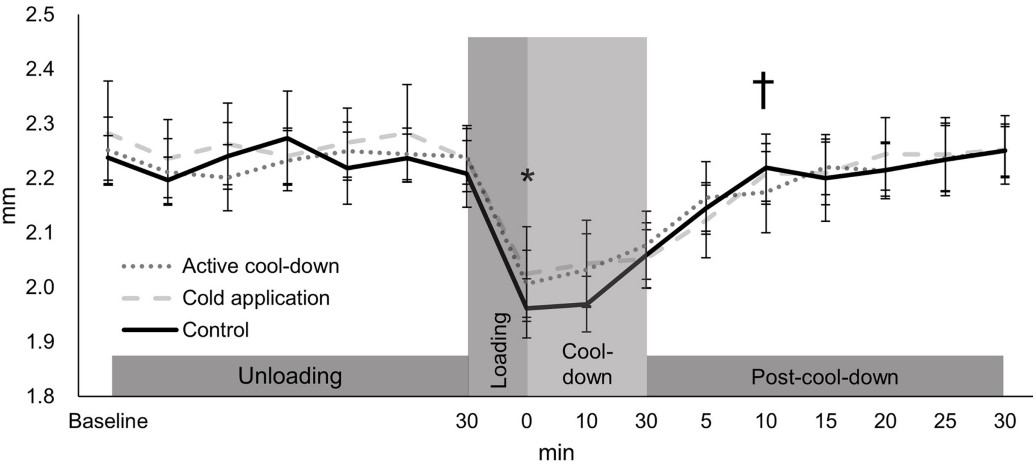

**Figure 3 Change in femoral cartilage intercondylar thickness among three conditions over time.** Values are mean and 95% confidence intervals. The asterisk and cross marks represent the time effect (condition collapsed): *Different from the baseline at 30-min (−11%, $p < 0.0001$, ES = 1.37). † Not different from baseline at 30-min ($p = 0.09$).

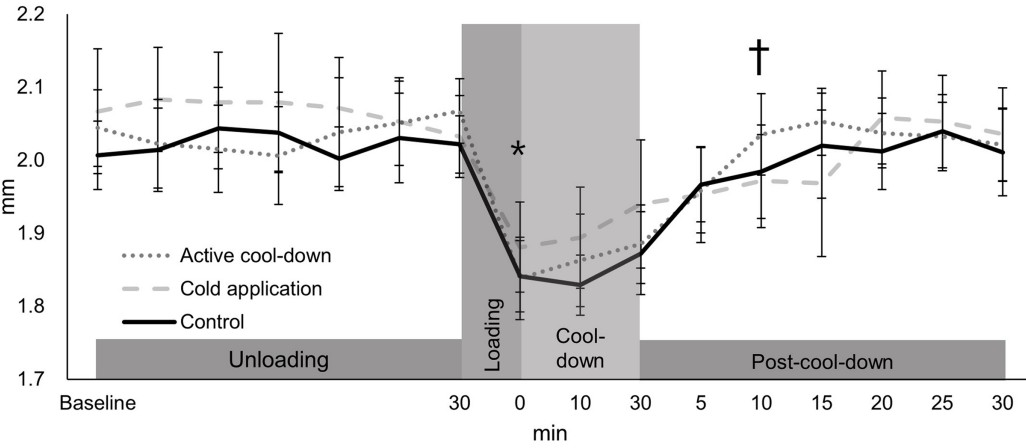

**Figure 4 Change in femoral cartilage lateral thickness among three conditions over time.** Values are mean and 95% confidence intervals. The asterisk and cross marks represent the time effect (condition collapsed): *Different from baseline at 30-min (−10%, $p < 0.0001$, ES = 1.58). † Not different from baseline at 30-min ($p = 0.64$).

## Lateral thickness

There was no condition effect over time in the lateral thickness (condition × time: $F_{30,705} = 1.24$, $p = 0.18$, Fig. 4). Regardless of time (condition effect: $F_{2,705} = 7.19$, $p < 0.0001$), lateral thickness in the cold application was thicker than the control condition (2.01 vs. 1.98 mm, $p = 0.0005$, ES = 0.24). Regardless of condition (time effect: $F_{15,705} = 26.32$, $p < 0.0001$, Fig. 4), there was an immediate reduction in the lateral thickness after running (1.85 mm, −10%, $p < 0.0001$, ES = 1.58), followed by a return to baseline levels (2.04 mm) at post-cool-down-10-min (1.9 mm, $p = 0.64$).

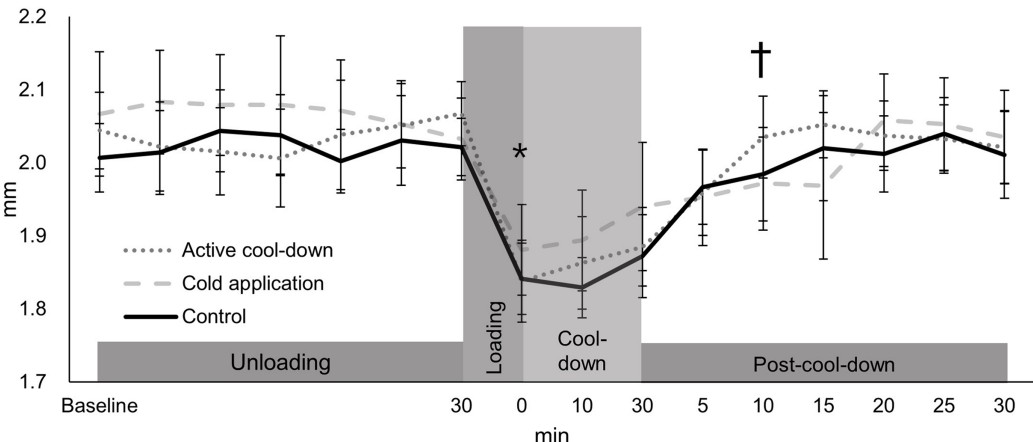

**Figure 5 Change in femoral cartilage medial thickness among three conditions over time.** Values are mean and 95% confidence intervals. The asterisk and cross marks represent the time effect (condition collapsed): * Different from baseline at 30-min (−8%, $p < 0.0001$, ES = 0.81). † Not different from baseline at 30-min ($p = 0.26$).

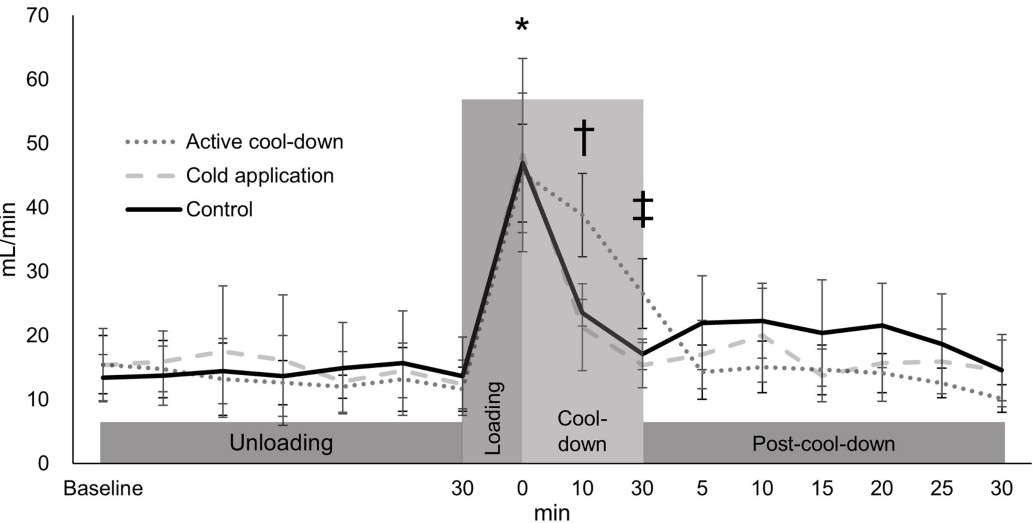

**Figure 6 Changes of blood flow volume of posteior tibial artery conditon among three conditions over time.** Values are mean and 95% confidence intervals. *Different from baseline at 30-min (all conditions: averaged of 316%, $p < 0.0001$, ES = 1.81). † Different from baseline at 30-min in the cold application and control condition ($p > 0.62$). ‡ Different from baseline at 30-min in the active cool-down condition ($p = 0.41$).

## Medial thickness

There was no condition effect over time in the medial thickness (condition × time: $F_{30,705} = 0.49$, $p = 0.99$, Fig. 5). Regardless of time (condition: $F_{2,705} = 3.24$, $p = 0.04$), medial thickness in the cold application was thicker than the control condition (1.86 *vs.* 1.84 mm, $p = 0.03$, ES = 0.16). Regardless of condition (time effect: $F_{15,705} = 10.31$, $p < 0.0001$, Fig. 5), there was an immediate reduction in the medial thickness after running (1.74 mm,

−8%, $p < 0.0001$, ES = 0.81), followed by a return to baseline levels (1.87 mm) at cool-down-5-min (1.82 mm, $p = 0.26$).

### Blood flow volume

Blood flow volume differed among three conditions over time (condition × time: $F_{30,705} = 2.36$, $p < 0.0001$, condition effect: $F_{2,705} = 1.59$, $p = 0.2$, Fig. 6). An increase in blood flow volume for all conditions was observed after running (average of 46.8 mL/min, 316%, $p < 0.0001$, ES = 1.81). In the active cool-down condition, the increased blood flow volume induced by running was maintained for 30-min (at cool-down-30-min (26.6 mL/min, 41%, $p < 0.0001$, ES = 1.64)) and returned to baseline levels at post-cool-down at 5-min (15.5 mL/min, $p = 0.41$). In the cold application (21.3 mL/min, 56%, $p < 0.0001$, ES = 1.13) and control (23.6 mL/min, 50%, $p < 0.0001$, ES = 1.47) conditions, blood flow volume returned to baseline levels (cold application: 15.4 mL/min, $p = 0.67$; control: 17.1 mL/min, $p = 0.62$) at cool-down-10-min.

## DISCUSSION

The purpose of this study was to examine the effects of three different cool-down protocols (active cool-down, cold application, and seated rest) on the recovery of femoral cartilage deformation following 30-min of treadmill running. Our hypotheses, that the recovery of running-induced cartilage deformation for the active cool-down and cold application conditions would be faster than the control condition, were not supported. The statistical condition effects with small ES (<0.24) should be interpreted as negligeable changes. The statistical time effects are indicative of the typical responses of articular cartilage to weight bearing-activity: The 30-min treadmill run consistently imposed similar physical demands (158.3 ± 26.4 bpm), step counts (4,731 ± 693 steps), and femoral cartilage deformation (8–11%) across sessions. Our results contribute to the existing evidence regarding the immediate response of femoral cartilage to running, consistent with previous data (*Harkey et al., 2017*; *Lee et al., 2023*; *Lim et al., 2024*). More importantly, running-induced femoral cartilage deformation required 40-min to return to pre-exercise levels, regardless of cool-down protocol. This observation suggests a practical implication: time could be a primary contributing factor to femoral cartilage recovery.

There are three known mechanisms of cartilage recovery: (1) Free swelling, involving fluid movement within the joint space to the exposed cartilage surface with non-contact area with a zero pressure (*Voinier et al., 2022*); (2) Passive swelling, where fluid moves from the unloaded contact area with a zero-pressure at the articular cartilage surface (*Voinier et al., 2022*); and (3) Tribological rehydration, where the load pushes fluid into articular cartilage during sliding under constant loading (*Voinier et al., 2022*). Free swelling and tribological rehydration, associated with cyclical movements such as walking and stretching, could have occurred during the active cool-down. Since fluid recovery *via* free swelling and tribological rehydration are known to be faster than passive swelling (*Voinier et al., 2022*), faster recovery in the active cool-down condition was expected. Additionally, there was a higher circulatory effect (confirmed by blood flow volume: see Fig. 6) in this condition. Regarding the cold application condition, recovery of femoral cartilage could

have mostly attributed to passive swelling associated with the seated position. Due to the relationship between tissue temperature and perfusion (*Esteves, McDonald & González-Alonso, 2024*), the use of ice bag would have accelerated the recovery process. However, neither the active cool-down nor cold application showed a better recovery rate, relative to the control condition. There could have been similar fluid recovery rates regardless of condition. Our results demonstrated that the recovery of running-induced femoral cartilage deformation is not influenced by post-exercise modalities. A previous study reported 30-min of recovery for a 5% reduction in femoral cartilage thickness after walking (*Harkey et al., 2018*). In combination with the previous results, our data (40-min of recovery for an average of 9.5% reduction) suggest a proportional relationship between deformation magnitude and recovery period. While circulation (*Kim & Park, 2020*) and temperature (*Lim et al., 2022*) are commonly considered when evaluating the level of recovery following exercise, coaches, athletes, and sports medicine practitioners should consider the time-dependent cartilaginous recovery process when implementing post-exercise recovery strategies.

Femoral cartilage response and recovery under the active cool-down and cold application conditions did not differ from those observed under the control condition. Our hypotheses on the effects of both interventions were according to the flow-dependent mechanism (*Sophia Fox, Bedi & Rodeo, 2009*). However, these indirect effects—such as an increase in the reabsorption rate involved with increased interstitial fluid—were not observed. Self-paced walking helped participants maintain the increased blood flow volume, which could have been attributed to a higher metabolic rate relative to other conditions, (*Seethapathi & Srinivasan, 2015*) for the demanding circulation to the working muscles (*Goto et al., 2007*). Increased blood flow volume could have been related to a larger diameter of the artery and/or a faster velocity of blood flow (*Blanco, 2015*). Although a greater body fluid delivery associated with the increased blood flow volume is assumed for the active-cool down at cool-down-10-min (Fig. 6), it did not affect cartilage recovery. Regarding the cooling intervention, it can be assumed that there was a temperature reduction in femoral cartilage based on previous focal knee joint cooling data (*Lee et al., 2017*; *Park, Song & Lee, 2022*). Previously, the metabolism (*Kocaoglu et al., 2011*) and function (*June & Fyhire, 2010*) of articular cartilage related to cold exposure have been studied, but its immediate effects on the recovery of exercise-induced deformation are relatively unknown. The role of blood flow increase or temperature reduction as contributing factors to recovery of exercise-induced deformation seems to be minimal. Our data suggest that body positioning, such as weight-bearing, static, and cyclic movement, might be more important than blood flow or temperature for the recovery of exercise-induced cartilage deformation.

There are several assumptions and limitations in our study. Although femoral cartilage deformation after running is in line with previous studies using ultrasonography (*Harkey et al., 2017*; *Lee et al., 2023*; *Lim et al., 2024*), it should be noted that a recent systematic review (*Khan et al., 2022*) on magnetic resonance image findings after running reported uncertainty in immediate cartilage deformation. We asked participants to keep their habitual diet, hydration status, and physical activity throughout the data collection period;

thus, we assume that morphological response to mechanical loading from treadmill running within each participant between sessions was not different. Similarly, the crossover study design could have controlled between-subject variability in running-induced exertion. The ultrasonographic image acquisition technique has limited access to the anterior femoral cartilage (*Akkaya et al., 2013*). While the sizes of femoral cartilage (see total length in Fig. 2) might vary between participants, our results are limited to an evaluation of the anterior portion, not the entire femoral cartilage. While there are known sex-specific differences in cartilage volume and surface area (*Faber et al., 2021*), males were selectively tested in this study. Additionally, the characteristics of the participant population (recreationally active healthy young individuals), exercise mode (treadmill running at a constant speed without cutting, hopping, landing, *etc.*), and environmental conditions (ambient temperature: 25 °C and relative humidity: 61%) should also be considered when applying our results to the field.

In summary, 30-min of treadmill running at a moderate intensity induced femoral cartilage deformation (8–11%). After running, a period of 30- to 40-min with an unloading position (seated with knees extended) was required for the recovery of femoral cartilage, and the use of common cool-down strategies (*e.g.*, active cool-down and cold application) did not enhance the recovery rate. Recovery of exercise-induced femoral cartilage deformation does not seem to be associated with blood flow volume and temperature reduction but a certain period of unloading.

### Funding

This work was supported by the Ministry of Education of the Republic of Korea and the National Research Foundation of Korea (NRF-2021S1A5A2A01062062). The funders had no role in study design, data collection and analysis, decision to publish, or preparation of the manuscript.

### Grant Disclosures

The following grant information was disclosed by the authors:
Ministry of Education of the Republic of Korea.
National Research Foundation of Korea: NRF-2021S1A5A2A01062062.

### Competing Interests

The authors declare that they have no competing interests.

### Author Contributions

- Sanghyup Park conceived and designed the experiments, performed the experiments, analyzed the data, prepared figures and/or tables, authored or reviewed drafts of the article, and approved the final draft.
- Junhyeong Lim conceived and designed the experiments, performed the experiments, analyzed the data, authored or reviewed drafts of the article, and approved the final draft.

- Jinwoo Lee performed the experiments, authored or reviewed drafts of the article, and approved the final draft.
- Seonggyu Jeon performed the experiments, authored or reviewed drafts of the article, and approved the final draft.
- Jaewon Kim performed the experiments, authored or reviewed drafts of the article, and approved the final draft.
- Jihong Park conceived and designed the experiments, performed the experiments, analyzed the data, prepared figures and/or tables, authored or reviewed drafts of the article, and approved the final draft.

### Human Ethics

The following information was supplied relating to ethical approvals (*i.e.*, approving body and any reference numbers):

Kyung Hee University.

### Data Availability

The raw data are available in the Supplemental File.

### Supplemental Information

Supplemental information for this article can be found online at http://dx.doi.org/10.7717/peerj.18302#supplemental-information.

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
