# Peer review of "Acute responses and recovery in the femoral cartilage morphology following running and cool-down protocols"

_PeerJ, doi:10.7717/peerj.18302_

## Round 0.1 · original submission · Major Revisions

The authors present a well-written manuscript. However, the reviewers raise major concerns, which needs to be addressed.

The authors need to finalize the acknowledgements section

Please also present standard deviation for anthropometric parameters in the abstract.

·

Basic reporting

Although there are some typos English quality is acceptable. Typos that should be corrected and needed exact p values are highlighted in the attached file.
References are recent and sufficient.
Structure of the article and the figures are well designed.
Results are relevant to the hypothesis.

Experimental design

No comment

Validity of the findings

No comment

Reviewer 2 ·

Basic reporting

No comment.

Experimental design

No comment.

Validity of the findings

No comment.

Additional comments

Review

General comment: Thank you for the opportunity to review the proposed manuscript. The reviewer reads with interest the proposed manuscript which aimed to compare the immediate effects of two common post-exercise cool-down methods to a control condition on subsequent morphologic change in femoral cartilage and vascular response in the posterior tibial artery after running. The manuscript is well-written, and clear and the data are well-presented. However, I do have some concerns regarding cartilage thickness quantification and justifications of including some outcome variables (i.e., fatigue level and blood lactate concentration). Please see specific comments below.

Specific comments:
Abstract: Please check the p-value presented as <0.0 and correct the value.

Introduction
Line 25: Is 60-80% fluid content in articular cartilage at resting? Or after activities? Please specify.

Lines 49-50: As a reader, I am curious why the duration required for cartilage to return to its normal shape is important in terms of cartilage health. Does a longer time to recover worsen cartilage health in some ways? Or vice versa? Cartilage deformation changes cartilage metabolism that deteriorates articular cartilage, so sustained deformation could elicit deleterious changes in articular cartilage. Or any other reasons? I wouldn’t stretch too much on this if there are no previous findings that can support the idea, but the authors should explain better why the duration of cartilage recovery from its deformation after activities in a clinical perspective, which makes this current work more important.

Lines 53-58: I appreciate that the authors tried to link changes in blood flow and temperature to morphological recovery in cartilage. However, the link is still weak to convince readers. If you could elucidate Moeini’s work that you cited, the link between blood flow and temperature, and cartilage morphology can be enhanced.

Methods
Lines 101-103: Please justify why you chose 140º for the position of the assessment with citation.
Lines 108: How many assessors performed the assessment? Have you run the test-retest reliability between assessors, if multiple, or within an assessor, if single? Please report the results.
Line 119: I did not know the authors measured blood lactate concentration until this point. Please link the outcome of blood lactate concentration to your research question and hypothesis in the Introduction.
Line 136: How many assessors performed the stretching? If multiple, how did you control the variability?
Lines 150: Have you considered using a relatively new method to quantify cartilage thickness (https://pubmed.ncbi.nlm.nih.gov/36930954/)? Basically, they segmented the superior and inferior borders and measuring the Euclidian distance between points along the segmented boundaries. This method will allow us to measure the overall thickness of the region of interest instead of measuring one single line of the thickness. Please consider re-quantifying your values by using the method that I shared with you. If not, please justify why your method fits better with your hypothesis.

Discussion
Lines 221-223: I haven’t seen this hypothesis that included blood lactate concentration and fatigue perception in the introduction. Please make sure your hypothesis is in line throughout the paper.
Line 226: Please add SDs for physical demands, step counts, and cartilage deformation.
Line 255: Consider starting with a topic sentence demonstrating that both cool-down interventions were not different from the control condition. Then, readers know what you wanted to highlight in this paragraph.
Lines 272-290: After reading this paragraph, I still cannot understand why you included blood lactate concentration in the current work. Furthermore, your statement “which seems to be related to femoral cartilage recovery in terms of duration” cannot be supported by “Blood lactate concentration returned to the baseline level at cool-down-30-min, while the fatigue perception took an additional 15-min to return” without correlational analyses. If you want to include blood lactate concentrations in the current work, please provide information supporting potential relationships between the concentrations and cartilage deformation/recovery and link up with your hypotheses in the introduction. If not, consider removing information about blood lactate concentrations (And fatigue perception) in Methods, Results, and Discussion, which will have readers focus on the main findings.

Reviewer 3 ·

Basic reporting

Well written, clear English - except for a few minor points listed below. Sufficient background provided. Good article structure. Figures and Tables appropriate.

Experimental design

Research question well defined. High standard of investigation performed. Methods described in sufficient detail - except minor points raised below.

Validity of the findings

Data is robust, controlled and statistically sound.
Conclusions are well stated and linked to findings.

Additional comments

An interesting article that includes blood flow, lactate and fatigue with knee cartilage thickness measures before, immediately after a 30-min run and during and after 3 different recovery conditions. The study is well designed and relevant to many regarding what (if any) recovery methods should be used after running activities for knee joint health.
The loading phase, treadmill run, was constant for all participants, but how was this normalised to each person, i.e. at what percent of their aerobic capacity was this load? Perhaps include more detail of the training history of the participants – were they of a similar running background/performance?
Not all studies have shown decreases in cartilage thickness following 30 min of exercise – this could be mentioned in the Discussion.
Minor points:
Abstract: clear and well structured. Though check missing parts of the p value and a bracket on line 11 ‘p<0.0was’
Better to use participants than subjects throughout the manuscript
Line 76 – change to: subjects (plural)
Line 86 – why only males included?
Line 146 change to: returned
Line 155 - change to: aligned parallel
Line 189 – change to: medial not medical
Lines 201-204 – it is not clear at what time points the data in the brackets relate to e.g. 56%, 50%, then state returns to baseline levels.

---

## Round 0.2 · accepted · Accept

I congratulate the authors on the well improved manuscript. No further comments from me or the reviewers.

·

Basic reporting

No comment

Experimental design

No comment

Validity of the findings

No comment

Reviewer 2 ·

Basic reporting

The article is clear and unambiguous, and professional English is used throughout.

Experimental design

This article is an original primary research within Aims and Scope of the journal.

Validity of the findings

The findings are impactful and novel.

Additional comments

I appreciate that the authors addressed all my suggestions deliberately. I have no further comments. Thank you again for the opportunity to review the great work!